# A Split Face Comparative Study to Evaluate the Efficacy of 40% Pyruvic Acid vs. Microdermabrasion with 40% Pyruvic Acid on Biomechanical Skin Parameters in the Treatment of Acne Vulgaris

**DOI:** 10.3390/jcm11206079

**Published:** 2022-10-14

**Authors:** Monika Rusztowicz, Karolina Chilicka, Renata Szyguła, Wiktoria Odrzywołek, Antoniya Yanakieva, Binnaz Asanova, Sławomir Wilczyński

**Affiliations:** 1Department of Health Sciences, Institute of Health Sciences, University of Opole, 45-040 Opole, Poland; 2Department of Basic Biomedical Science, Faculty of Pharmaceutical Sciences in Sosnowiec, Medical University of Silesia in Katowice, 41-200 Sosnowiec, Poland; 3Department of HTA, Faculty of Public Health, Medical University of Sofia, 1427 Sofia, Bulgaria; 4Medical College Yordanka Filaretova, Medical University of Sofia, 1606 Sofia, Bulgaria

**Keywords:** acne vulgaris, microdermabrasion, pyruvic acid, moisturizing, sebum

## Abstract

The synergy of cosmetic acids, with their keratolytic and antibacterial properties, with the mechanical exfoliation of the epidermis brings faster and better treatment results. The aim of the study was to compare the effects of using only pyruvic acid and the synergy of microdermabrasion and chemical exfoliation. In total, 14 women diagnosed with acne took part in the study. Two areas were marked on the participants’ faces: the right side (microdermabrasion treatment and a preparation containing pyruvic acid 40%) and the left side (preparation containing pyruvic acid 40%) without mechanical exfoliation. A series of four treatments was performed at 2-week intervals. Skin parameters such as stratum corneum hydration and sebum secretion were measured. Before the treatments, all patients had moderate acne according to GAGS (Min: 19, Max: 22, Md: 20), and after the treatments, it decreased to mild acne according to GAGS (Min: 13, Max: 17, Md: 140). On the right side of the face, there was a statistically significant reduction in sebum secretion in all the examined areas of the face and increase in the hydration of the stratum corneum. On the left side of the face, the differences were also observed in the decrease of sebum value and increase of hydration level; however, they were smaller than on the right side. The use of microdermabrasion in combination with pyruvic acid led to better results in the case of increased hydration and reduction of sebum secretion than using only pyruvic acid treatment.

## 1. Introduction

Acne vulgaris is a dermatological disease that is characterized by whiteheads, blackheads, pustules, nodules, cysts or inflammatory papules [1]. The most common factors that can cause disease include: abnormal keratinization of the pilosebaceous canal, bacterial colonization (*Cutibacterium acnes*), hormonal disorders and excessive sebum production [2,3,4]. As a result of neglect or the acute course of the disease, scarring and discoloration of the skin may occur, which may be associated with a significantly reduced quality of life. Patients struggling with acne vulgaris often feel isolated, excluded from society and the disease limits their social life. An increased course of acne increases the risk of anxiety and depression [5,6,7,8].

Cosmetology offers many treatments, both with the use of chemicals and modern equipment that are able to alleviate the symptoms of acne. These include, among others, cosmetic acids, which cause coherence between keratinocytes and exfoliation of corneocytes. The advantages are also the reconstruction of the epidermis structure, as well as the reduction of sebum secretion onto the epidermis surface. The acids that are most often used in anti-acne treatments are: alpha hydroxy, beta hydroxy and alpha-keto acids [9,10,11].

Pyruvic acid is an alpha-keto acid that is found in apples, vinegar and fermented fruits, among others. It has good antibacterial and exfoliating properties, and also has a sebo-regulating effect. When it comes to acne scars, it also works well on this level, because it has the ability to stimulate the formation of new collagen and elastic fibers. Physiologically, PA converts to lactic acid, and it has a low risk of scarring. PA differs from lactic acid due to the presence of a ketone group in the alpha position instead of the hydroxyl group. PA is classified as a medium peeling agent that has a positive effect on acne vulgaris and greasy skin. It is also useful in photodamage of pigmentation in patients with light skin (tyrosinase inhibition properties) [12].

Microdermabrasion is classified as an apparatus treatment, which is a form of mechanical superficial exfoliation of the epidermis. It is a safe, non-invasive and effective method. This treatment is used for the reduction of acne, acne scars, photoaging, stretch marks and for superficial hyperpigmentation. Superficial exfoliation of the epidermis allows for better penetration of other substances that will be applied after the mechanical exfoliation treatment. As of today, there are three types of microdermabrasion: diamond microdermabrasion using diamond-coated heads, corundum microdermabrasion using corundum and oxybrasion using a stream of saline solution (0.9% NaCl). Thanks to special tips, diamond microdermabrasion can exfoliate the upper layers of the epidermis in a controlled way. The depth of exfoliation depends on the diamond-coated head used, the suction power set by the cosmetologist and the pressure of his hands on the skin of the person undergoing the procedure [13,14,15].

The aim of the study was to check whether the synergy of treatments would have a positive effect on the improvement of skin quality in people with acne vulgaris.

## 2. Materials and Methods

### 2.1. Patients

In total, 14 women with acne vulgaris were classified in this study. Full characteristics of the patients are summarized in Table 1. The Bioethicas Committee of the Opole Medical School agreed (KB/54/NOZ/2019) to conduct the project entitled: “Assessment of selected skin parameters and quality of life after cosmetics procedures in people with acne vulgaris and oily skin”.

#### 2.1.1. Inclusion Criteria

Inclusion criteria for this study were no dermatological treatment within 12 months, no current hormonal contraception, age 19–22 and mild-to-moderate acne which was measured by the global acne severity scale (GAGS). GAGS is clinical tool for assessing the severity of acne vulgaris that was first proposed by Doshi et al., in 1997.

#### 2.1.2. Exclusion Criteria

The study had contraindications that made it impossible for some people to participate. These included: taking oral medications within the last 3 months, taking isotretinoin within the last year, taking contraceptives, tendency to keloids, sun exposure weeks before and after procedure, telangiectasias, skin cancers, pregnancy and breastfeeding, viral, bacterial and fungal skin diseases, hypersensitivity to acids, skin irritation, active inflammation, active rosacea, psoriasis and atopic dermatitis.

Acne, excessive seborrhea, blackheads, whiteheads and papules were observed in the volunteers. Before and after the treatment series, the GAGS scale was used to determine the severity of acne and to check the effect of the treatments on the improvement of the skin condition of the patients. The scale includes the following areas: nose, cheeks, forehead, chin, as well as the back and chest. Each of them is assigned a number based on size: nose = 1; left cheek = 2; right cheek = 2; forehead = 2; chin = 1. Depending on the degree of severity, each lesion is given a grade: no cutaneous conditions—0, comedones—1, papules—2, pustules—3 and nodules—4. The local score calculated for each area has the formula: Local score = factor × Grade (0–4). The global score is composed of the sum of the local results: 1–18 = mild acne, 19–30 = moderate acne, 31–38 = severe acne, above 39 = acne with very severe course [16]. All volunteers underwent a series of four treatments applied every 2 weeks.

### 2.2. Procedure

The skin parameters were measured before and one month after the end of the treatment. The participants removed their face makeup the day before the measurements in the evening and did not apply any preparations to the face skin. Measurements were taken in the morning, and the test participants, after arriving in the room, acclimatized for about 30 minutes. The humidity in the room was 40–50%, while the temperature was 20–21 degrees C.

Corneometer CM 825 was used to measure stratum corneum hydration, while a Sebumeter SM 815 (Courage + Khazaka Electronic GmbH, Köln, Germany) was used to measure sebum. The measurement points were as follows: 1 cm above the brow, 1 cm from the lobe of the nose, cheek (5 cm from the lobe of the nose), 1 cm from the corner of the mouth. The measurements were made symmetrically.

Thermal camera FLIR T420 (FLIR Systems Company, Sweden) was used for thermal measurements. The thermal resolution was <0.045 °C; the wavelength range was 7.5–13 μm; the resolution of the obtained image was 320 × 240 pixels. FLIR ResearchIR software version 3.5 (FLIR^®®^ Systems, Inc., Wilsonville, OR, USA) was used for analysis of the data.

The study was conducted from March 2021 to April 2021.

Before starting the treatment procedure, the face was cleansed with micellar fluid. The treatment was divided into two parts: the right and the left side of the face. The entire face was first degreased with alcohol. On the right side of the patients’ face, microdermabrasion was performed for a period of 5 minutes (tip gradation 200, negative pressure 15 mmHg). Then, after the end of the treatment, the face was washed with hydrogen peroxide solution and a preparation containing 40% pyruvic acid (Perfarma Pyruvic Peeling 40 strong) was applied. The preparation was applied to the right surface of the face with a brush for 2 minutes and then washed off with water. No microdermabrasion was performed on the left side of the patients’ face. After degreasing the face, 40% pyruvic acid was applied with a brush to the left part of the face for 2 minutes and washed off with water. Then, a post-treatment cream with a 50+ filter (Dives Global Protection) was applied to the entire treatment area.

For home care, it was recommended to wash the face twice a day with the preparation Cetaphil MD Dermoprotector (Aqua, Glycerin, Hydrogenated Polyisobutene, Cetearyl Alcohol, Macadamia Ternifolia Seed Oil/Macadamia Ternifolia Nut Oil, Ceteareth-20, Tocopheryl Acetate, Dimethicone, Acrylates/C10-30 Alkyl Acrylate Crosspolymer, Benzyl Alcohol, Citric Acid, Farnesol, Panthenol, Phenoxyethanol, Sodium Hydroxide, Stearoxytrimethylsilane, Stearyl Alcohol, FIL 0133.V02.). After applying the above-mentioned preparation, the subjects were to apply Alantan Plus cream (20 mg allantoin and 50 mg dexpanthenol as a 50% solution of panthenol in propylene glycol, lanolin, liquid paraffin, cetostearyl alkohol, etyl parahydroxybenzoate, metyl parahydroxybenzoate, propyl parahydroxybenzoate, purified water, polawax).

The patients were instructed not to use a swimming pool or sauna, to avoid exposure to natural and artificial radiation, and to use photoprotection every day and not to use any preparations other than Cetaphil MD Dermoprotector and Alantan Plus cream.

During the entire series of treatments and a month after its treatment, it was forbidden to use any other cosmetic and aesthetic medicine treatments. Oral supplementation with preparations that could reduce the amount of sebum produced was also forbidden.

### 2.3. Statistical Analysis

Statistica 13.3 software was used for the statistical analysis of the results. Wilcoxon test was used to compare the effects before and after the treatments. The results at the level of *p* < 0.05 were considered statistically significant.

## 3. Results

The use of the treatments had a statistically significant effect on the severity of acne in each of the patients (*p* < 0.001) (Figure 1). Before the treatments, all patients had moderate acne according to GAGS (Min: 19, Max: 22, Md: 20), and after the treatments, it decreased to mild acne according to GAGS (Min: 13, Max: 17, Md: 14) (Table 2).

On the right side of the face, where treatments using both microdermabrasion and 40% pyruvic acid were applied, there was a statistically significant reduction in sebum secretion in all the examined areas of the face (*p* < 0.001) (Figure 2, Table 3). The reduction in sebum secretion on the right side of the face occurred in all patients, and the median sebum value decreased from 77.5 ug/cm^2^ on the cheek to 113.0 ug/cm^2^ on the forehead.

On the left side of the face, where only the 40% pyruvic acid procedure was performed, a reduction in sebum secretion was also achieved, but the effects were smaller than on the right side (Figure 2, Table 3). On the left side of the face, near the forehead, the reduction in sebum secretion did not fully reach the level of statistical significance (*p* = 0.055); in the remaining areas, the differences were statistically significant, and the median sebum secretion value decreased by o 7.5 ug/cm^2^, respectively, (*p* < 0.01) on the nose, by 10 µg/cm^2^ (*p* < 0.05) around the corner of the mouth and by 11.5 ug/cm^2^ (*p* < 0.01) on the cheek.

The microdermabrasion procedure and the application of 40% pyruvic acid on the right side of the face resulted in a statistically significant increase in the hydration of the stratum corneum in the forehead (*p* < 0.01), the corner of the mouth (*p* < 0.01) and the cheek (*p* < 0.05) (Figure 3, Table 4). The median value of skin hydration increased by 7.2 in the forehead area, 2.3 in the nasal lobe, 3.2 in the corner of the mouth and 4.9 in the cheek area.

On the left side of the face, after the treatment with 40% pyruvic acid, improvement in skin hydration could also be observed, but it did not reach statistical significance in any of the examined areas of the face (Figure 3, Table 4).

The left and right side of the face were compared in the thermography assessment (Figure 4). Thermographic images of the face were recorded before and after treatment. The analysis of the patient’s face temperature (Region of Interest) before and after procedure did not show statistically significant differences both on the right and on the left side of the face.

## 4. Discussion

Acne vulgaris therapy should lead to reduced skin lesions, inflammation and sebum secretion as well as to improved appearance [17]. Various treatments, e.g., topical therapies, oral antibiotic treatment, isotretinoin and hormonal therapies can cause side effects. Chemical peeling is one of the most popular non-invasive cosmetic treatments [18,19]. The use of chemical peels in combination with microdermabrasion for the reduction of acne lesion can be a response to the search for effective and safe methods of acne treatment [20].

The effectiveness of pyruvic acid has been confirmed in numerous studies. In the study by Jaffary et al., patients underwent four treatments with 50% hydroalcoholic solution of pyruvic acid for 2-week intervals. Patients showed a statistically significant reduction in the number of comedones, papules and ASI [21]. Marczyk et al. observed a statistically significant decrease in the level of secreted sebum after the third application (out of six) of 50% pyruvic peel [22]. The findings of Chilicka et al.’s research indicate a significant reduction of acne lesion after the PA six peeling sessions at 2-week intervals. An effect of reducing skin greasiness was also observed [12].

Microdermabrasion removes the superficial epidermal layer, which may contribute to the changes of the hydrolipid barrier. Fąk et al. examine the changes in hydration and sebum level on the skin after microdermabrasion. Thirty minutes after treatment on the cheeks and immediately after on the T-zone, significant differences in stratum corneum hydration have been observed. The reduction of sebum secretion level was observed immediately after the treatment [23].

The above studies show the effectiveness of pyruvic acid and microdermabrasion in reducing sebum secretion and increasing the hydration of the epidermis. The improvement of these skin parameters can reduce acne lesions, and the combination of these two methods may lead to better results than their individual action.

In our study, the use of microdermabrasion in combination with 40% pyruvic acid contributed to the reduction of sebum secretion in each of the measured areas. The differences were greater than when using the pyruvic acid procedure without microdermabrasion. The value of hydration of the stratum corneum increased after a series of treatments with pyruvic acid and microdermabrasion than in the case of treatments without microdermabrasion. Higher hydration of the stratum corneum and greater reduction of sebum secretion after the use of a series of microdermabrasion treatments in combination with pyruvic acid may be the result of the synergic effect of the combination of these treatments on acne-prone skin. Microdermabrasion can lead to exfoliation of epidermal cells and can accelerate and increase the penetration of pyruvic acid. Pyruvic acid has the ability to transform into a component of NMF (Natural Moisturizing Factor) lactic acid which leads to better hydration of the epidermis.

## 5. Study Limitations

The research sample can be seen to be limited. In the future, we would like to increase the number of patients, as well as expand the group to include men, not only women.

## 6. Conclusions

In conclusion, after the four peeling sessions using pyruvic acid and pyruvic acid with microdermabrasion, all patients showed better skin parameters in terms of reduced sebum secretion and increased stratum corneum hydration. However, the use of microdermabrasion in combination with pyruvic acid led to better results with increased hydration and reduction of sebum secretion than using only the pyruvic acid treatment. It should be remembered that dermatological treatment cannot be replaced by cosmetological treatment.

## Figures and Tables

**Figure 1 jcm-11-06079-f001:**
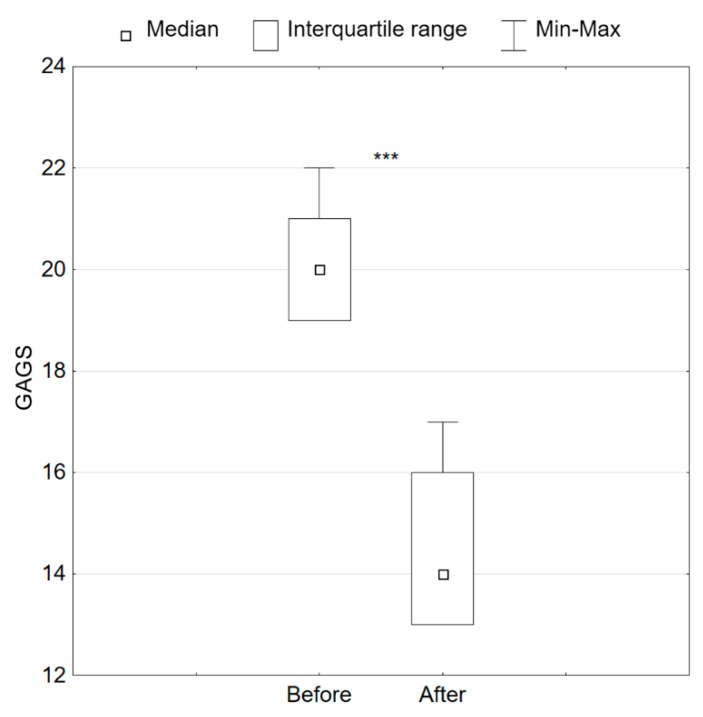
Advancement of acne according to GAGS before and after treatments, *** *p* < 0.001.

**Figure 2 jcm-11-06079-f002:**
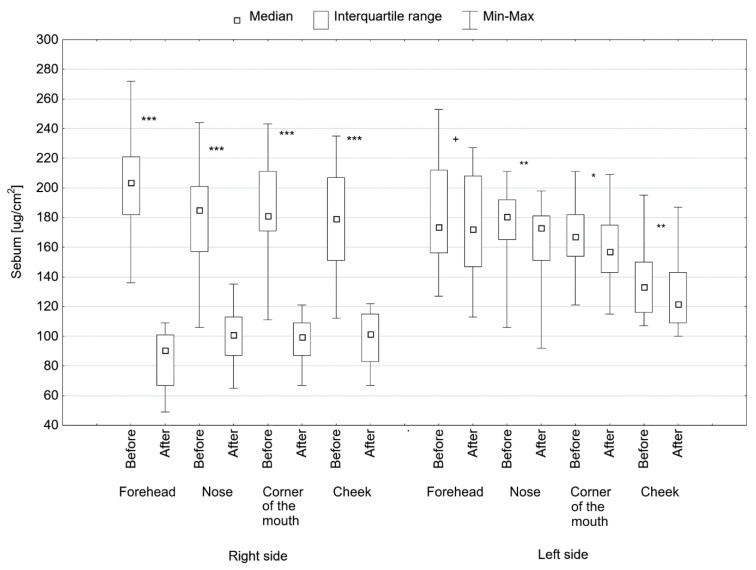
Sebum value in various areas of the face on the right side, treated with microdermabrasion and pyruvic acid, and on the left side, treated only with acid before and after treatments, ^+^
*p* = 0.055, * *p* < 0.05, ** *p* < 0.01, *** *p* < 0.001.

**Figure 3 jcm-11-06079-f003:**
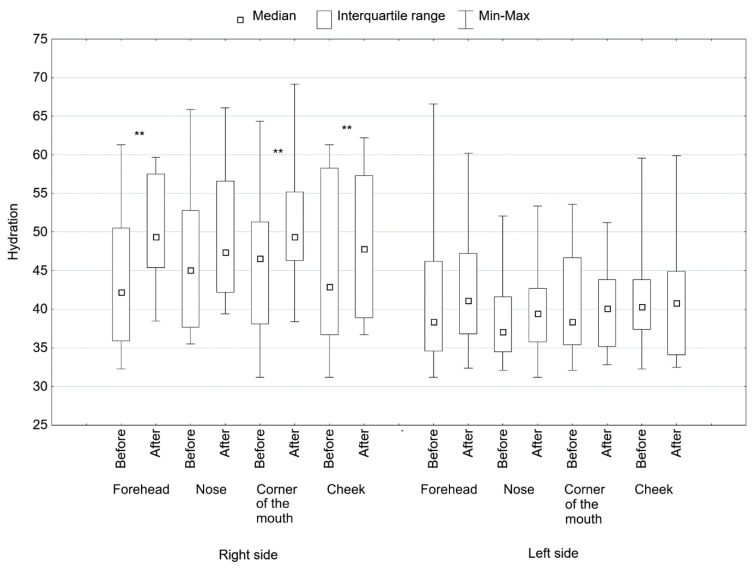
Moisturizing the skin in various areas of the face on the right side, treated with microdermabrasion and pyruvic acid, and on the left side, treated only with acid before and after treatments, ** *p* < 0.01.

**Figure 4 jcm-11-06079-f004:**
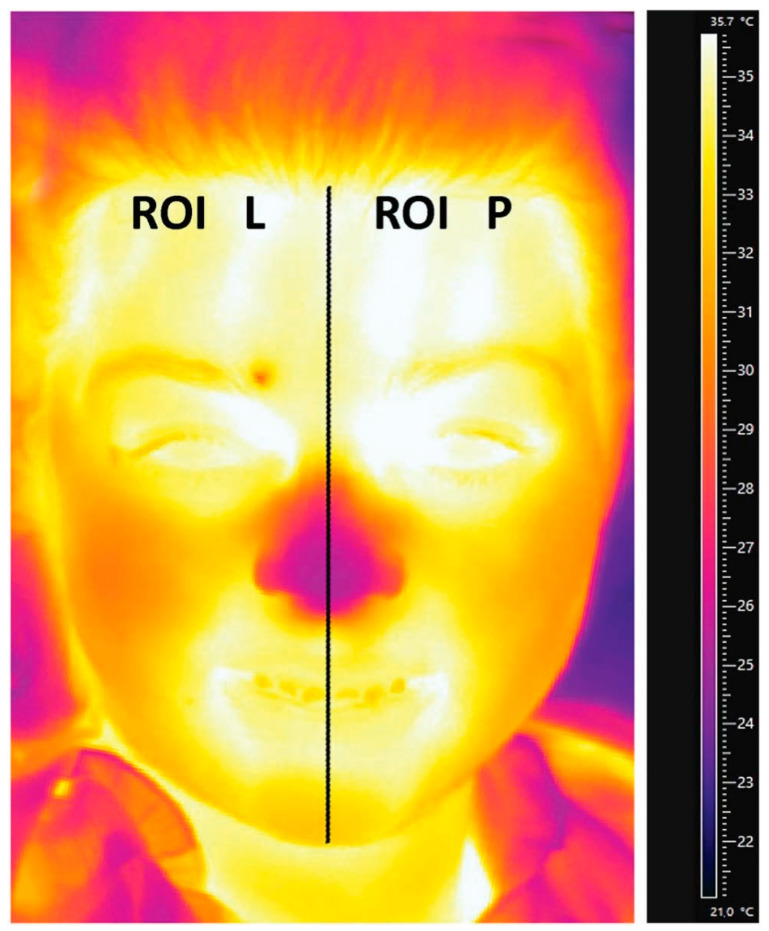
A thermogram of a sample patient with the left and right treatment area marked. The mean temperature difference in the ROI L and ROI R areas for all the studied patients was not statistically significant.

**Table 1 jcm-11-06079-t001:** Patient characteristics.

	N	Mean	SD	Min	Max
age (years)	14	19.6	0.7	19.0	21.0
duration of the disease (years)	14	5.4	1.7	2.0	8.0

**Table 2 jcm-11-06079-t002:** Advancement of acne according to GAGS before and after treatments, *** *p* < 0.001, Md—Median, Q1—1st quartile, Q3—3rd quartile, Min—Minimum, Max—Maximum.

	GAGS ***	
	Before	After
Md	20	14
Q1	19	13
Q3	21	16
Min	19	13
Max	22	17

**Table 3 jcm-11-06079-t003:** Sebum value in various areas of the face on the right side, treated with microdermabrasion and pyruvic acid, and on the left side, treated only with acid before and after treatments, ^+^
*p* = 0.055, * *p* < 0.05, ** *p* < 0.01, *** *p* < 0.001, Md—Median, Q1—1st quartile, Q3—3rd quartile, Min—Minimum, Max—Maximum.

	Right Side								Left Side							
	Forehead ***		Nose ***		Corner of the Mouth ***		Cheek ***		Forehead ^+^		Nose **		Corner of the Mouth *		Cheek **	
	Before	After	Before	After	Before	After	Before	After	Before	After	Before	After	Before	After	Before	After
Md	203.5	90.5	185.0	101.0	181.0	99.5	179.0	101.5	173.5	172.0	180.5	173.0	167.0	157.0	133.0	121.5
Q1	182.0	67.0	157.0	87.0	171.0	87.0	151.0	83.0	156.0	147.0	165.0	151.0	154.0	143.0	116.0	109.0
Q3	221.0	101.0	201.0	113.0	211.0	109.0	207.0	115.0	212.0	208.0	192.0	181.0	182.0	175.0	150.0	143.0
Min	136.0	49.0	106.0	65.0	111.0	67.0	112.0	67.0	127.0	113.0	106.0	92.0	121.0	115.0	107.0	100.0
Max	272.0	109.0	244.0	135.0	243.0	121.0	235.0	122.0	253.0	227.0	211.0	198.0	211.0	209.0	195.0	187.0

**Table 4 jcm-11-06079-t004:** Moisturizing the skin in various areas of the face on the right side, treated with microdermabrasion and pyruvic acid, and on the left side, treated only with acid before and after treatments, ** *p* < 0.01, Md—Median, Q1—1st quartile, Q3—3rd quartile, Min—Minimum, Max—Maximum.

	Right Side								Left Side							
	Forehead **		Nose		Corner of ** the Mouth		Cheek **		Forehead		Nose		Corner of the Mouth		Cheek	
	Before	After	Before	After	Before	After	Before	After	Before	After	Before	After	Before	After	Before	After
Md	42.2	49.4	45.1	47.4	46.6	49.4	42.9	47.8	38.4	41.2	37.1	39.5	38.4	40.1	40.3	40.8
Q1	35.9	45.4	37.7	42.2	38.1	46.3	36.7	38.9	34.6	36.8	34.5	35.8	35.4	35.2	37.4	34.1
Q3	50.5	57.5	52.8	56.6	51.3	55.2	58.3	57.3	46.2	47.2	41.6	42.7	46.7	43.8	43.8	44.9
Min	32.3	38.5	35.5	39.4	31.2	38.4	31.2	36.7	31.2	32.4	32.1	31.2	32.1	32.8	32.3	32.5
Max	61.3	59.7	65.9	66.1	64.4	69.2	61.3	62.2	66.6	60.2	52.1	53.4	53.6	51.2	59.6	59.9

## Data Availability

The data presented in this study are available on request from the corresponding author.

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
