# Peer review of "A Split Face Comparative Study to Evaluate the Efficacy of 40% Pyruvic Acid vs. Microdermabrasion with 40% Pyruvic Acid on Biomechanical Skin Parameters in the Treatment of Acne Vulgaris"

_jcm, 2022, doi:10.3390/jcm11206079_

Round 1
Reviewer 1 Report
1-introduction- although I have presented the problem well delimited, I have as a suggestion: insert the justification of the study in the last paragraph
2-line 135: write in english
3-hydrogen peroxide: concentration?
4-Did you perform the sample calculation? could you present?
Author Response
Response to Reviewer 1
Dear Editor and Reviewer
We would like to thank the Editor and Reviewer for their time and valuable comments. Your suggested edits and constructive criticism were invaluable, and we hope that this version of the manuscript will be better. We sincerely appreciate your suggestions and did our best to incorporate them into the revised manuscript. Our responses are highlighted in red.
English language and style
(x) English language and style are fine/minor spell check required
- We checked all manuscript.
Comments and Suggestions for Authors
1-introduction- although I have presented the problem well delimited, I have as a suggestion: insert the justification of the study in the last paragraph
- We added this sentence.
2-line 135: write in English
- We changed it.
3-hydrogen peroxide: concentration?
4-Did you perform the sample calculation? could you present?
- solution

Reviewer 2 Report
Probably instead of "The synergy of cosmetic acids, which keratolytic and antibacterial properties" on line 14 you would like to say something else, maybe "The synergy of cosmetic acids, with their keratolytic and antibacterial properties"
On line 33, instead of "Acne vulgaris is a dermatological disease in characterized by whiteheads", maybe you would like to have something else like "Acne vulgaris is a dermatological disease that is characterized by whiteheads"
On lines 39-40, instead of "the disease causes limit their social life" maybe you would like to have something like "the disease limits their social life"
On lines 40-41, instead of "An increased course of acne increases the risk anxiety and depression" you might have "An increased course of acne increases the risk of anxiety and depression"
On line 56, is there a real word "pigmentatory disorders"? Maybe pigmentation or pigmentary?...
On line 58, instead of "Microdermabrasion is classified as apparatus treatment" maybe "Microdermabrasion is classified as an apparatus treatment"
On line 81, instead of "globalacne severityscale" maybe "global acne severity scale"
On line 110, I think that instead of ”above the brow” would be easier to understand ”above the forehead”
On lines 135-136... well, you got me ”Po zastosowaniu wyżej wy-135 mienionego preparatu probantki miaÅ‚y nakÅ‚adać krem Alantan Plus”
On line 161, probably is better to use the complete form of words ”Fig. 2, Tab. 3”. Same on line 166, 184, 188
On line 163, please correct ”77.5 ug/cm2” and ”113,0 ug/cm2” using proper letters. Same on 169 - ”7,5 ug/cm2”, ”p <0 01”, on line 170 - ”10 μg / cm2”, ”11,5 ug/cm2”
In Tables 3 and 4, should be the forehead, not forhead
On line 208, instead of ”isotreretinoin” please have ”isotretinoin”
The number of subjects is very low, so the whole statistics might be irrelevant. You should mention that...
The bibliography is rather short...
Author Response
Response to Reviewer 2
Dear Editor and Reviewer
We would like to thank the Editor and Reviewer for their time and valuable comments. Your suggested edits and constructive criticism were invaluable, and we hope that this version of the manuscript will be better. We sincerely appreciate your suggestions and did our best to incorporate them into the revised manuscript. Our responses are highlighted in red.
English language and style
(x) Moderate English changes required
- We checked all manuscript.
Comments and Suggestions for Authors
Probably instead of "The synergy of cosmetic acids, which keratolytic and antibacterial properties" on line 14 you would like to say something else, maybe "The synergy of cosmetic acids, with their keratolytic and antibacterial properties"
On line 33, instead of "Acne vulgaris is a dermatological disease in characterized by whiteheads", maybe you would like to have something else like "Acne vulgaris is a dermatological disease that is characterized by whiteheads"
On lines 39-40, instead of "the disease causes limit their social life" maybe you would like to have something like "the disease limits their social life"
On lines 40-41, instead of "An increased course of acne increases the risk anxiety and depression" you might have "An increased course of acne increases the risk of anxiety and depression"
On line 56, is there a real word "pigmentatory disorders"? Maybe pigmentation or pigmentary?...
On line 58, instead of "Microdermabrasion is classified as apparatus treatment" maybe "Microdermabrasion is classified as an apparatus treatment"
On line 81, instead of "globalacne severityscale" maybe "global acne severity scale"
On line 110, I think that instead of ”above the brow” would be easier to understand ”above the forehead”
- We will leave the brow- thank You.
On lines 135-136... well, you got me ”Po zastosowaniu wyżej wy-135 mienionego preparatu probantki miaÅ‚y nakÅ‚adać krem Alantan Plus”
On line 161, probably is better to use the complete form of words ”Fig. 2, Tab. 3”. Same on line 166, 184, 188
On line 163, please correct ”77.5 ug/cm2” and ”113,0 ug/cm2” using proper letters. Same on 169 - ”7,5 ug/cm2”, ”p <0 01”, on line 170 - ”10 μg / cm2”, ”11,5 ug/cm2”
In Tables 3 and 4, should be the forehead, not forhead
On line 208, instead of ”isotreretinoin” please have ”isotretinoin”
The number of subjects is very low, so the whole statistics might be irrelevant. You should mention that...
The bibliography is rather short...
- We added the study limitations to the article.

Round 2
Reviewer 2 Report
There are still some minor escapes, in lines 157, 171, 177, and 178 you might change "," with "." in various numbers (ex 11,5 should be 11.5)
In table 4, please write forehead instead of forhead
Author Response
Dear Reviewer,
We changed all suggestions.